# Moving Frame Net: SE(3)-Equivariant Network for Volumes

**Mateus Sangalli**                                                    MATEUS.SANGALLI@MINESPARIS.PSL.EU
**Samy Blusseau**                                                         SAMY.BLUSSEAU@MINESPARIS.PSL.EU
**Santiago Velasco-Forero**                                           SANTIAGO.VELASCO@MINESPARIS.PSL.EU
**Jesús Angulo**                                                          JESUS.ANGULO@MINESPARIS.PSL.EU
*Mines Paris, PSL University, Centre for Mathematical Morphology, France*

**Editors:** Sophia Sanborn, Christian Shewmake, Simone Azeglio, Arianna Di Bernardo, Nina Miolane

## Abstract

Equivariance of neural networks to transformations helps to improve their performance and reduce generalization error in computer vision tasks, as they apply to datasets presenting symmetries (e.g. scalings, rotations, translations). The method of moving frames is classical for deriving operators invariant to the action of a Lie group in a manifold. Recently, a rotation and translation equivariant neural network for image data was proposed based on the moving frames approach. In this paper we significantly improve that approach by reducing the computation of moving frames to only one, at the input stage, instead of repeated computations at each layer. The equivariance of the resulting architecture is proved theoretically and we build a rotation and translation equivariant neural network to process volumes, i.e. signals on the 3D space. Our trained model overperforms the benchmarks in the medical volume classification of most of the tested datasets from MedMNIST3D.

**Keywords:** Lie groups, Group equivariance, 3D image classification, Moving frames

## 1. Introduction

There is currently a great interest in building machine learning methods that respect symmetries, such as translation, rotation and other physical gauge symmetries. Convolutional neural networks (CNN) are translation equivariant neural networks that have shown great success in a wide variety of tasks related to image processing and understanding. Recent work has shown that designing group-equivariant CNNs that exploit additional symmetries via group convolutions has even further increased their performance (Cohen and Welling, 2016; Worrall et al., 2017; Weiler et al., 2018; Cohen et al., 2019; Bogatskiy et al., 2020).

The method of moving frames (Cartan, 1935; Fels and Olver, 1999), initially proposed by Élie Cartan to produce differential invariants, was recently applied to define SE(2)-equivariant convolutional neural networks, i.e. CNNs equivariant to rotations and translations in 2D, by Sangalli et al. (2022). The network constructed with differential invariants provides an alternative to group convolution when constructing group equivariant networks. In this work, we build on the recent work of Sangalli et al. (2022) to develop a SE(3)-equivariant neural network. We propose a new approach that uses differential invariants on 3D CNNs by computing a moving frame from the input of a neural network and applying it to Gaussian $n$-jets in order to obtain equivariant architectures for SE(3). This technique is leveraged to propose a novel CNN architecture that we call *SE(3)-Moving Frame Network* (SE3MovFNet) [1]. The new architecture can be applied to volumetric data. Empirically, we

---

1. its implementation can be found at https://github.com/mateussangalli/MovingFrameNetwork

show that SE3MovFNet improves the performance of competitive CNNs on a collection of datasets for 3D medical image classification problems.

The paper is organized as follows: We discuss related work in the literature in Section 2 and we introduce concepts of group equivariance, as well as some basics on the method of moving frames, in Section 3. In Section 4 we derive a moving frame in SE(3) and show how one can obtain an equivariant neural network architecture from a moving frame and in Section 5 we introduce the SE3MovFNet based on the moving frames method In Section 6 we validate de SE3MovFNet in a task of medical volume classification and overperform most of the benchmarks. We end the paper with concluding remarks in Section 7.

## 2. Related Work

In the literature of group-equivariant networks there exist many approaches to plane rotation-equivariant networks, for example, Cohen and Welling (2016); Worrall et al. (2017); Weiler et al. (2018). Also on 2D rotation-equivariant networks, some approaches are based on differential operators (Shen et al., 2020; Jenner and Weiler, 2022; Sangalli et al., 2022). In particular, the current approach is an extension of the moving frames-based SE(2)-equivariant neural network in Sangalli et al. (2022). In the domain of 3D CNNs, two of the possible data representations are point clouds and volumetric data. Many approaches that seek equivariance to space rotations are for CNNs that process point cloud data (Thomas et al., 2018; Chen et al., 2021; Melnyk et al., 2021; Thomas, 2020).

Our work focuses on defining SE(3)-equivariant networks for data based on voxels, i.e., volumetric data. Some other approaches that aim to achieve this result are: Worrall and Brostow (2018) achieves equivariance to a discrete subgroup of SO(3); Weiler et al. (2018) uses a steerable filter basis based on spherical harmonics to learn general SE(3)-equivariant filters and Shen et al. (2022) does the same thing using filters based on partial differential operators. Our approach uses differential operators like Shen et al. (2022) but instead of using a steerable filter basis we apply a moving frame to invariantize the network, which consists of evaluating each neighborhood with a rotation computed at the first layer.

## 3. Technical Background

In this section we introduce both the concepts of group action and equivariance and the basic concepts behind the method of moving frames. The final goal of the paper is to propose a class of group-equivariant networks based on the method of moving frames.

### 3.1. Group Actions and Equivariance

Given a group $\mathcal{G}$ and a set $\mathcal{M}$, a *group action*[2] of $\mathcal{G}$ on $\mathcal{M}$ is a map $\pi : \mathcal{G} \times \mathcal{M} \to \mathcal{M}$ such that $\forall \mathbf{x} \in \mathcal{M}$ $\pi(e, \mathbf{x}) = \mathbf{x}$ where $e$ is the neutral element of $\mathcal{G}$ and $\forall g, h \in \mathcal{G}$, $\forall \mathbf{x} \in \mathcal{M}$ $\pi(g, \pi(h, \mathbf{x})) = \pi(g \cdot h, \mathbf{x})$. We denote for all $g \in \mathcal{G}$, $\mathbf{x} \in \mathcal{M}$, $g \cdot \mathbf{x} := \pi(g, \mathbf{x})$. If $\mathcal{G}$ is a Lie group, $\mathcal{M}$ a smooth manifold and $\pi$ is a smooth map, then $\pi$ is a Lie group action. See Appendix A for properties of group actions. In this paper we have an implicit assumption of locality of the group action. A *local Lie group action* is a smooth map $\pi : U \to \mathcal{M}$ where

---

2. Here we deal only with *left group actions* but *right group actions* can have analogous results.

$U \subseteq \mathcal{G} \times \mathcal{M}$ is an open set such that $\{e\} \times \mathcal{M} \subseteq U$, satisfying $\forall \mathbf{x} \in \mathcal{M}$, $\pi(e, \mathbf{x}) = \mathbf{x}$ and $\forall g, h \in \mathcal{G}, \mathbf{x} \in \mathcal{M}$ s.t. $\{(h, \mathbf{x}), (g, \pi(h, \mathbf{x})), (g \cdot h, \mathbf{x})\} \subseteq U$, we have $\pi(g, \pi(h, \mathbf{x})) = \pi(g \cdot h, \mathbf{x})$. We use the same notation as group actions for local group actions.

Given sets $\mathcal{M}$ and $\mathcal{N}$ acted upon by $\mathcal{G}$ an operator $\phi : \mathcal{M} \to \mathcal{N}$ is *equivariant* if $\forall g \in \mathcal{G}, x \in \mathcal{M}$, $\phi(g \cdot \mathbf{x}) = g \cdot \phi(\mathbf{x})$. We assume that the action on $\mathcal{M}$ is not the identity to avoid trivial cases. *Invariance* is a special case of equivariance where the action on $\mathcal{N}$ is the identity, i.e. $\phi(g \cdot \mathbf{x}) = \phi(\mathbf{x})$.

Given manifolds $X$ and $Y$, when $\mathcal{G}$ acts on $X$ with actions we define the action on the space of smooth functions $C^\infty(X, Y)$ as, for all $f \in C^\infty(X, Y)$, $\mathbf{x} \in X$, $g \in \mathcal{G}$

$$(g \cdot f)(\mathbf{x}) := f(g^{-1} \cdot \mathbf{x}). \tag{1}$$

As this paper is focused on exploring equivariant networks on signals, this is the type of action we seek equivariance to. Since we are interested in rotations in the input domain we do not consider actions that change the output of the function $f$, but the general method presented in this paper is capable of dealing with that.

### 3.2. The Method of Moving Frames

**Moving Frames.** Let $\mathcal{M}$ be an $m$-dimensional smooth manifold and $\mathcal{G}$ be an $r$-dimensional Lie group that acts on $\mathcal{M}$. A *moving frame* (Fels and Olver, 1999) is a $\mathcal{G}$-equivariant map $\rho : \mathcal{M} \to \mathcal{G}$ which in particular satisfies, $\forall \mathbf{z} \in \mathcal{M}$, $g \in \mathcal{G}$

$$\rho(g \cdot \mathbf{z}) = \rho(\mathbf{z}) \cdot g^{-1}. \tag{2}$$

A moving frame $\rho$ induces the function $\mathbf{z} \mapsto \rho(\mathbf{z}) \cdot \mathbf{z}$ which is constant over each orbit $\mathcal{O}_{\mathbf{z}} = \{g \cdot \mathbf{z}, g \in \mathcal{G}\}$. Namely, $\forall \mathbf{z} \in \mathcal{M}$, $g \in \mathcal{G}$, $\rho(g \cdot \mathbf{z}) \cdot g \cdot \mathbf{z} = \rho(\mathbf{z}) \cdot \mathbf{z}$.

**Invariantization.** The main interest of having a moving frame from the perspective of equivariant deep learning is the *invariantization* it defines. Given an operator $F : \mathcal{M} \to \mathcal{N}$, its invariantization is defined as $\forall \mathbf{z} \in \mathcal{M}$, $\imath[F](\mathbf{z}) := F(\rho(\mathbf{z}) \cdot \mathbf{z})$. The invariantization of an operator is invariant with respect to the group action as $\imath[F](g \cdot \mathbf{z}) = \imath[F](\mathbf{z})$ for every $\mathbf{z} \in \mathcal{M}, g \in \mathcal{G}$. Applying the invariantization to an invariant operator returns the same operator, therefore the set of invariant operators is the set of invariantized operators.

In our case, objects of interest (volumes, images, etc) are functions $f : X \to Y$ where $X = \mathbb{R}^p$ and $Y = \mathbb{R}^q$, $p, q \in \mathbb{N}^*$. They can be modeled as submanifolds of the manifold $\mathcal{M} = X \times Y$ by identifying them by their graph where each point has coordinates $(\mathbf{x}, u) = (\mathbf{x}, f(\mathbf{x}))$. In that case, if we can decompose the action of $\mathcal{G}$ into an action on $X$ and an action on $Y$, then we can associate each invariant operator on $\mathcal{M}$ to an equivariant one on the space of functions $f : X \to Y$ (see appendix B). We use this framework in this paper.

**Cross-Section.** A *cross-section* to the group orbits is a submanifold $K \subseteq \mathcal{M}$ of dimension complementary to the group dimension i.e. $\dim K = m - r$ that intersects each orbit transversally[3]. If the intersection happens at most once it is a regular cross-section. If $\mathcal{G}$ acts freely and regularly on $\mathcal{M}$ and given a regular cross-section $K$ to the group orbits, then for each $z \in \mathcal{M}$ there is a unique element $g_z \in \mathcal{G}$ such that $g_z \cdot \mathbf{z} \in K$. The function $\rho : \mathcal{M} \to \mathcal{G}$ mapping each $z$ to $g_z$ is a moving frame (Fels and Olver, 1999; Olver, 2007).

---

3. The tangent spaces of $K$ and of the orbit $\mathcal{O}_z$ span the tangent space of $\mathcal{M}$ at the intersection $K \cap \mathcal{O}_z$.

**Jet-Bundle.** The $n$-th order jet bundle (Olver, 1993), or *jet-space*, $J^n(\mathcal{M})$ is an extension of a manifold $\mathcal{M}$ given by equivalence classes of functions. For us the jet bundle is particularly useful when the group action is not free on $\mathcal{M}$, as prolonging the manifold to a sufficiently high-order jet bundle and extending the group action to this space can result in a free action, enabling the definition of a moving frame.

In this section we define the jet-space for spaces of the form $\mathcal{M} = X \times Y$ where $X = \mathbb{R}^p$, $Y = \mathbb{R}$. Given a multi-index $I = (i_1, \cdots, i_p) \in \mathbb{N}^p$, we will note $|I| = \sum_{k=1}^p i_k$ its modulus, and let us denote by $\mathcal{I}_n^p = \{I \in \mathbb{N}^p, |I| \leq n\}$ the set of multi-indices in $\mathbb{N}^p$ of modulus at most $n \in \mathbb{N}$. For $\mathbf{x} \in X$ and $n \in \mathbb{N}$, we introduce $\mathcal{J}_{\mathbf{x}}^{(n)} f$ the operator mapping $C^\infty(X,Y)$ to $Y_p^{(n)} = \mathbb{R}^{\binom{n+p}{n}}$, and defined for any $f \in C^\infty(X,Y)$ by $\mathcal{J}_{\mathbf{x}}^{(n)}(f) = \left(\partial^I f(\mathbf{x})\right)_{I \in \mathcal{I}_n^p} = \left(\partial^{i_1}\partial^{i_2}\ldots\partial^{i_p} f(\mathbf{x})\right)_{(i_1,\ldots,i_p)\in\mathcal{I}_n^p}$. Then for any $u^{(n)} = \left(u_I\right)_{I\in\mathcal{I}_n^p} \in Y_p^{(n)}$ and any $\mathbf{x} \in X$, $(\mathcal{J}_{\mathbf{x}}^{(n)})^{-1}(u^{(n)})$ is an equivalence class for the equivalence relation $f_1 \sim f_2 \iff \mathcal{J}_{\mathbf{x}}^{(n)}(f_1) = \mathcal{J}_{\mathbf{x}}^{(n)}(f_2)$. This class is represented in particular by the polynomial function defined for any $\mathbf{t} \in X$ by

$$u(\mathbf{t}) = \sum_{I=(i_1,\ldots,i_p)\in\mathcal{I}_n^p} \frac{u_I}{I!}(t_1 - x_1)^{i_1}\ldots(t_p - x_p)^{i_p}, \tag{3}$$

with $I! = i_1! i_2! \ldots i_p!$. It is the Taylor polynomial of order $n$ at $\mathbf{x}$ of any function of the class. The $n$th-order jet space of $\mathcal{M}$, noted $J^n(\mathcal{M})$, is the union of all such equivalence classes, and can therefore be indentified to $X \times Y_p^{(n)}$. According to the above, for an element $(\mathbf{x}, u^{(n)}) = (\mathbf{x}, (u_I)_{I\in\mathcal{I}_n^p})$, each $u_I$ is also a partial derivative of $u$ evaluated in $\mathbf{x}$, namely $u_I = \partial^I u(\mathbf{x})$. For example, if $p = 3$ and $\mathbf{x} = (x, y, z)$, $u_{(0,0,0)} = u(\mathbf{x})$, $u_{(1,0,0)} = \frac{\partial u}{\partial x}(\mathbf{x}) = u_x(\mathbf{x})$, $u_{(1,1,0)} = \frac{\partial^2 u}{\partial x \partial y}(\mathbf{x}) = u_{xy}(\mathbf{x})$ and so on. In practice we will often use these partial derivative notations to identify elements of the jet space, and omit the variable as it is explicit from the first component. For example in the case $p = 3$, $n = 2$, an element $\mathbf{z} \in \mathcal{M} = J^0(\mathcal{M})$ is identified by $\mathbf{z} = (\mathbf{x}, u) = (x, y, z, u)$ and an element $\mathbf{z}^{(2)} \in J^2(\mathcal{M})$ by $\mathbf{z}^{(2)} = (\mathbf{x}, u^{(2)}) = (x, y, z, u, u_x, u_y, u_z, u_{xx}, u_{xy}, u_{yy}, u_{xz}, u_{yz}, u_{zz})$.

**Prolongation of the Group Action.** Because smooth functions at a point $\mathbf{x}$ can be associated to an element of the jet-space and vice-versa, it makes sense that the actions on functions induces an action in the jet-space. This action is computed by associating an $n$-jet to a function and computing its derivatives at the transformed point $g \cdot \mathbf{x}$. Formally, given a point $(\mathbf{x}, u^{(n)}) \in J^n(\mathcal{M})$, let $u \in C^\infty(X, Y)$ be a function such that $u^{(n)} = \mathcal{J}_{\mathbf{x}}^{(n)} u$, without loss of generality we can choose $u$ to be the polynomial (3). We define the *prolongation* of the action of $\mathcal{G}$ on $\mathcal{M}$ to the jet-space $J^n(\mathcal{M})$, given by, for $g \in \mathcal{G}$ and $\mathbf{z}^{(n)} = (\mathbf{x}, u^{(n)}) \in J^n(\mathcal{M})$

$$g \cdot \mathbf{z}^{(n)} = g \cdot (\mathbf{x}, u^{(n)}) \coloneqq (g \cdot \mathbf{x}, \mathcal{J}_{g\cdot\mathbf{x}}^{(n)}(g \cdot u)). \tag{4}$$

The expression (4) is well defined as it can be verified that it does not depend on the choice of $u$ Olver (1993) and is a group action. The intuition behind evaluating the derivatives at the point $g \cdot \mathbf{x}$, is that the value in $u$ at $\mathbf{x}$ is the same as the value of $g \cdot u$ at $g \cdot \mathbf{x}$, i.e. $(g \cdot u)(g \cdot \mathbf{x}) = u(g^{-1} \cdot g \cdot \mathbf{x}) = u(\mathbf{x})$, however its $n$-jet is not the same (see Figure 1).

**Fundamental invariants.** Invariantizations of operators in the jet-space are referred to as differential invariants and the invariants[4] $\imath[x_i]\, \imath[u_I]$, $1 \leq 1 \leq p$, $0 \leq |I| \leq n$ are called

---

4. Here we abuse notation and denote $\imath[x_i]$ as the invariantization of the projection $(\mathbf{x}, u^{(n)}) \mapsto x_i$ and we denote $\imath[u_I]$ the invariantization of the projection $(\mathbf{x}, u^{(n)}) \mapsto u_I$.

*fundamental invariants* because every differential invariant of order $n$ can be expressed as a functional combination $F(\imath[x_1], \ldots, \imath[x_p], (\imath[u_I])_{0 \le |I| \le n})$ (Olver, 2007). Conversely every function of the fundamental invariants is a differential invariant.

## 4. Moving Frame and Differential Invariants for SE(3) on Volumes

The group SE(3) of rotations and translations in dimension three, is the semi-direct product of Lie groups SO(3) (rotations) and $\mathbb{R}^3$ (translations) and since both are 3-dimensional, then SE(3) is a 6-dimensional Lie group. Here a *volume* refers to a signal on a 3-dimensional Euclidean domain, i.e. functions of the type $f : \mathbb{R}^3 \to \mathbb{R}^q$.

We derive SE(3)-equivariant operators on volumes using the method of moving frames. Volumes are represented as submanifolds of $\mathcal{M} = \mathbb{R}^3 \times \mathbb{R}$. We consider the case $q = 1$, but keeping in mind that for higher dimensions it is just a matter of channel-wise application. SE(3) acts on $\mathcal{M}$ by rotating and translating the spatial coordinates $\mathbf{x}$, i.e.

$$\forall (R, \mathbf{v}) \in SE(3), \forall (\mathbf{x}, u) \in \mathcal{M}, \quad \pi_{R,\mathbf{v}}(\mathbf{x}, u) = (R \cdot \mathbf{x} + \mathbf{v}, u), \tag{5}$$

If we proceed to extend $\mathcal{M}$ to the first-order jet space we will find that SE(3) does not act freely on $J^1(\mathcal{M})$. Indeed, the orbit of a point $(\mathbf{x}, u, u_x, u_y, u_z)$ is the Cartesian product of $\mathbb{R}^3$, $\{u\}$ and a sphere with radius $\sqrt{u_x^2 + u_y^2 + u_z^2}$, hence it has dimension $5 \neq \dim SE(3) = 6$. Therefore it is necessary to prolong the action to the second order jet-space in order to be able to obtain a moving frame. In this section we use a matrix notation for compactness: we denote $\nabla u = [u_x, u_y, u_z]^T$ and

$$Hu = \begin{bmatrix} u_{xx} & u_{xy} & u_{xz} \\ u_{xy} & u_{yy} & u_{yz} \\ u_{xz} & u_{yz} & u_{zz} \end{bmatrix}. \tag{6}$$

In that way, the coordinates of the second order jet-space are identified by $\mathbf{z}^{(2)} = (\mathbf{x}, u^{(2)}) = (\mathbf{x}, u, \nabla u, Hu)$.

We compute the prolonged action following (4). Choosing some $u \in C^\infty(X, Y)$ such that $u^{(n)} = \mathcal{J}_{\mathbf{x}}^{(n)} u$ the action becomes $(R, \mathbf{v}) \cdot (\mathbf{x}, u^{(n)}) = (R\mathbf{x} + \mathbf{v}, \mathcal{J}_{R\mathbf{x}+\mathbf{v}}^{(n)}((R, \mathbf{v}) \cdot u))$. In order to compute the second order jet-space, we compute the gradient and Hessian matrix of the function $\tilde{u}(\mathbf{t}) = ((R, \mathbf{v}) \cdot u)(\mathbf{t})$ at the point $\tilde{\mathbf{x}} = R\mathbf{x} + \mathbf{v}$. From (1) we have $\tilde{u}(\mathbf{t}) = u((R, \mathbf{v})^{-1} \cdot \mathbf{t}) = u(R^T(\mathbf{t} - \mathbf{v}))$, thus applying the chain rule we have $\nabla \tilde{u}(\mathbf{t}) = R \nabla u(R^T(\mathbf{t} - \mathbf{v}))$ and substituting $\mathbf{t}$ by $\tilde{\mathbf{x}}$ we have $\nabla \tilde{u}(\tilde{\mathbf{x}}) = R \nabla u(\mathbf{x})$. An example illustrating this in the two-dimensional case is shown in Figure 1. A similar argument can be applied to show that $H\tilde{u}(\tilde{\mathbf{x}}) = R Hu(\mathbf{x}) R^T$.

$$\pi_{R,\mathbf{v}}(\mathbf{z}^{(2)}) = \pi_{R,\mathbf{v}}(\mathbf{x}, u, \nabla u, Hu) = (R\mathbf{x} + \mathbf{v}, u, R\nabla u, R Hu R^T). \tag{7}$$

A similar reasoning can be applied to describe the coordinates of the higher order jet-spaces as symmetric tensors and obtain the prolongation of the group action of higher order using tensor contraction. This action decomposes into an action on $X$ and an action on $Y^{(n)}$.

Now we can find a cross-section that will give us a moving frame. Because the cross-section has to have a dimension complementary to the group, we use six equations to

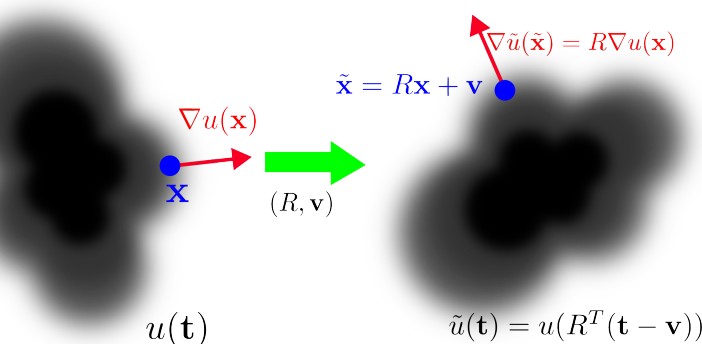

Figure 1: Example of the first order prolonged action of SE(2). The gradient and transformed gradients evaluated at $\mathbf{x}$ and $\tilde{\mathbf{x}} = (R, \mathbf{v}) \cdot \mathbf{x}$ are related by the rotation $R$, thus the first order action is also a rotation by $R$. In the second order case, the same effect is observed on the eigenvalues of the Hessian matrix.

construct it. We also add some inequalities to ensure regularity. We propose $K = \{\mathbf{x} = 0, u_{xy} = u_{xz} = u_{yz} = 0, u_{xx} \geq u_{yy} \geq u_{zz}, u_x > 0, u_y > 0\}$ i.e. $\mathbf{x} = 0$ and $Hu$ is diagonal with its diagonal sorted in non-increasing order. $Hu$ is symmetric so we can find a $P \in SO(3)$[5] such that $PHuP^T$ is diagonal, resulting in the moving frame $\rho : (\mathbf{x}, u) \in \mathcal{M} \mapsto (P, -P\mathbf{x}) \in \mathcal{G}$.

### 4.1. Equivariant Network from a Fixed Moving Frame

From the prolonged group action (7) we can deduce that the non-trivial fundamental invariants of the jet-space of order two are $u$, $v_i^T \cdot \nabla u$ and $\lambda_i$, $i = 1, 2, 3$, where the $v_i$s are the eigenvectors of $Hu$ (columns of $P$) and the $\lambda_i$s are the eigenvalues of $Hu$ (diagonal coefficients of $PHuP^T$).

Following the approach of Sangalli et al. (2022), a two-layer SE(3)-equivariant neural network can be obtained using invariants of order two as follows:

1. Let the volume $f \in C^\infty(X, Y) = C^\infty(\mathbb{R}^3, \mathbb{R})$ be the input to the network. First we compute for all $\mathbf{x}$, $\mathcal{J}_\mathbf{x}^{(2)} f = (f(\mathbf{x}), \nabla f(\mathbf{x}), Hf(\mathbf{x}))$, followed by the computation of the fundamental invariants of order 2, which we will denote $\imath[\mathbf{z}_\mathbf{x}^0] = \rho(\mathbf{x}, \mathcal{J}_\mathbf{x}^{(2)} f) \cdot (\mathbf{x}, \mathcal{J}_\mathbf{x}^{(2)} f) = \rho(\mathbf{z}_\mathbf{x}^0) \cdot \mathbf{z}_\mathbf{x}^0$ where $\mathbf{z}_\mathbf{x}^0 = (\mathbf{x}, \mathcal{J}_\mathbf{x}^{(2)} f)$. Let $\psi_1 : J^n(\mathcal{M}) \to Y_1$, where $Y_1 = \mathbb{R}^q$, be a smooth map. In a deep learning context we assume $\psi_1$ to be a multilayer perceptron (MLP). The first layer $\phi_1 : C^\infty(\mathbb{R}^3, Y) \to C^\infty(\mathbb{R}^3, Y_1)$, $Y_1 = \mathbb{R}^q$, is given by

$$\phi_1[f](\mathbf{x}) = \psi_1(\imath[\mathbf{z}_\mathbf{x}^0]) = \psi_1(\rho(\mathbf{z}_\mathbf{x}^0) \cdot \mathbf{z}_\mathbf{x}^0) = \imath[\psi_1](\mathbf{z}_\mathbf{x}^0) \tag{8}$$

and it is SE(3)-equivariant, because $\imath[\psi_1]$ is an invariant applied as in Appendix B.

---

5. if $\det P = -1$ we can multiply one of its rows by $-1$ so that the new matrix has determinant 1.

2. We build the second layer analogously. The output of the first layer is a signal $f_1 = (f_1^1, f_1^2, \ldots, f_1^q) = \phi_1(f) : X \to Y_1$. We compute the derivatives $\mathcal{J}_{1,\mathbf{x}}^{(2)} f = (\mathcal{J}_{\mathbf{x}}^{(2)}(f_1^1), \ldots, \mathcal{J}_{\mathbf{x}}^{(2)}(f_1^q))$ and the fundamental invariants $(\imath[\mathbf{z}_{\mathbf{x}}^{1,1}], \cdots, \imath[\mathbf{z}_{\mathbf{x}}^{1,q}])$ where $\mathbf{z}_{\mathbf{x}}^{1,j} = (\mathbf{x}, \mathcal{J}_{\mathbf{x}}^{(2)}(f_1^j))$ and $\imath[\mathbf{z}_{\mathbf{x}}^{1,j}] = \rho(\mathbf{z}_{\mathbf{x}}^{1,j}) \cdot \mathbf{z}_{\mathbf{x}}^{1,j}$. Let $\psi_2 : (J^n(\mathcal{M}))^q \to Y_2$ be a function given by an MLP. The second layer can be computed from the output of the first by

$$\phi_2'[f_1](\mathbf{x}) = \psi_2\Big(\rho(\mathbf{z}_{\mathbf{x}}^{1,1}) \cdot \mathbf{z}_{\mathbf{x}}^{1,1}, \cdots, \rho(z_{\mathbf{x}}^{1,q}) \cdot z_{\mathbf{x}}^{1,q}\Big). \tag{9}$$

Again this function is equivariant because it is a function of invariants. The second layer $\phi_2 = \phi_2' \circ \phi_1$ is equivariant because it is a composition of equivariant operators. This process can be repeated to obtain $L$ equivariant layers $\phi_l = \phi_l' \circ \phi_{l-1}$, $0 \leq l \leq L$.

Using the cross-section $K$, the approach described above requires the computation of the gradient of eigenvectors and eigenvalues with respect to the matrix entries, and even when using a closed polynomial expression to write these values, it can be quite challenging numerically. With both the numerical or closed form expression of the eigenvectors, the training of the networks resulted in exploding gradients in our early experimentation. We propose a new solution which limits the computations to only one moving frame.

The alternative we propose is the following. Instead of computing the differential invariants at each layer, involving the computation of the moving frame based on the previous layer's feature maps, we compute the moving frame based only on the network input signal and compute all subsequent layers based on this moving frame.

Computing a two-layer network as in the previous example, $f_1$ is obtained exactly as in step 1. Now from $f_1$ we compute $\mathcal{J}_{1,\mathbf{x}}^{(2)} f$ for all $\mathbf{x}$. Given some $\psi_2 : (J^n(\mathcal{M}))^q \to \mathbb{R}^{q'}$ (which again should be regarded as an MLP) we can obtain the output of the second layer. In contrast to (9), however, we transform $\rho(\mathbf{z}_{\mathbf{x}}^{1,j})$ according to $\rho(\mathbf{z}_{\mathbf{x}}^0) = \rho(\mathbf{x}, \mathcal{J}_{\mathbf{x}}^{(2)} f)$, not to itself obtaining

$$\phi_2'(f_1)(\mathbf{x}) = \psi_2(\rho(\mathbf{z}_{\mathbf{x}}^0) \cdot \mathbf{z}_{\mathbf{x}}^{1,1}, \ldots, \rho(\mathbf{z}_{\mathbf{x}}^0) \cdot \mathbf{z}_{\mathbf{x}}^{1,q}). \tag{10}$$

The next result shows that repeated application of (10) defines a SE(3)-equivariant network.

**Proposition 1** *Let $X = \mathbb{R}^3$, $Y = \mathbb{R}^{q_0} = \mathbb{R}$ and $Y_l = \mathbb{R}^{q_l}$ for $1 \leq l \leq L$, assume that SE(3) acts on $X \times Y_l$ like (5). Let $\rho : X \times Y^{(n)} \to SE(3)$ be a moving frame. Let smooth maps $\psi_l \in C^\infty(J^n(\mathcal{M})^{q_{l-1}}, Y_l)$ for $1 \leq l \leq L$. The functions $\phi_l : C^\infty(X,Y) \to C^\infty(X,Y_l)$, $1 \leq l \leq L$ defined by, for all $f \in C^\infty(X,Y)$, $\mathbf{x} \in X$, denoting $\mathbf{z}_{\mathbf{x}}^0 = (\mathbf{x}, \mathcal{J}_{\mathbf{x}}^{(n)} f)$ and $\mathbf{z}_{\mathbf{x}}^l = (\mathbf{x}, \mathcal{J}_{\mathbf{x}}^{(n)} \phi_l(f)_j)_{0 \leq j \leq q_l}$, $1 \leq l \leq L$,*

$$\phi_1[f](\mathbf{x}) = \psi_1\left(\rho(\mathbf{z}_{\mathbf{x}}^0) \cdot \mathbf{z}_{\mathbf{x}}^0\right) \tag{11}$$

*and, for $1 < l \leq L$ either*

$$\phi_l[f](\mathbf{x}) = \psi_l\left(\rho(\mathbf{z}_{\mathbf{x}}^0) \cdot \mathbf{z}_{\mathbf{x}}^{l-1}\right) \tag{12}$$

*or (assuming $q_l = q_{l-1}$)*

$$\phi_l[f](\mathbf{x}) = \phi_{l-1}[f](\mathbf{x}) + \psi_l\left(\rho(\mathbf{z}_{\mathbf{x}}^0) \cdot \mathbf{z}_{\mathbf{x}}^{l-1}\right) \tag{13}$$

*are SE(3)-equivariant for all $1 \leq l \leq L$. Where the of $\mathcal{G}$ on $J^n(\mathcal{M})^{q_l}$, $1 \leq l \leq L$, is the action on $J^n(\mathcal{M})$ applied coordinate-wise.*

**Proof** See Appendix C.                                                         ∎

Because the moving frame is fixed we do not have to compute the gradients of an eigen decomposition, as the moving frame can be seen as an input to the network we only have to compute the gradients of an eigen decomposition once, and the moving frame can be seen as an input to the network. Indeed the expressions $\rho(\mathbf{z}_\mathbf{x}^0) \cdot \mathcal{J}_\mathbf{x}^{(n)} \phi_l[f]$ are linear with respect to the values of of the $l$-th layer $\phi_l[f]$.

## 5. Moving Frame Nets for SE(3) Acting on Volumes

### 5.1. Gaussian Derivatives

In order to compute the differential invariants we use Gaussian derivatives. Gaussian derivatives have already been used in neural networks to produce structured receptive fields (Jacobsen et al., 2016; Penaud-Polge et al., 2022; Sangalli et al., 2022) in CNNs. Gaussian derivatives are used to compute the derivatives of a Gaussian filtered volume $f : \Omega \to \mathbb{R}$ defined on a grid $\Omega \subseteq \mathbb{R}^3$:

$$\frac{\partial^{i+j+k}}{\partial x^i \partial y^j \partial z^k}(f * G_\sigma) = f * \frac{\partial^{i+j+k}}{\partial x^i \partial y^j \partial z^k} G_\sigma = f * G_\sigma^{i,j,k}. \tag{14}$$

We refer to the Gaussian $n$-jet of a volume as the Gaussian derivatives of order $\leq n$ $f_\sigma^{(n)} := (f * G_\sigma^{i,j,k})_{0 \leq i+j+k \leq n}$. We can also identify the Gaussian $n$-jet by tensor coordinates. In particular for $n = 2$ we write $f_\sigma^{(n)} = (f_\sigma, \nabla_\sigma f, H_\sigma f)$ where $f_\sigma = f * G_\sigma$, $\nabla_\sigma$ is the Gaussian gradient i.e. Gaussian derivatives of order one and $H_\sigma f$ is the Gaussian Hessian, i.e. Gaussian derivatives of order two. Given a orthogonal matrix for each point $P : \Omega \to SO(3)$ (e.g. the matrices defining a moving frame) we denote the local prolonged action by $(P \cdot f_\sigma^{(n)})(\mathbf{x})$, e.g. for $n = 2$ $(P \cdot f_\sigma^{(n)})(\mathbf{x}) = (f_\sigma(\mathbf{x}), P(\mathbf{x})\nabla_\sigma f(\mathbf{x}), P(\mathbf{x})H_\sigma f(\mathbf{x})P(\mathbf{x})^T)$. If $f : \Omega \to \mathbb{R}^q$ is a multi-channel volume we can apply these operations channel-wise.

Gaussian filters are already rotation-equivariant, so their composition with a differential invariant yields a rotation-equivariant operator. Moreover, they avoid the issues inherent with discrete signals and reduce the negative impact sampling signals. These properties motivate the use of Gaussian $n$-jets to compute the invariants.

### 5.2. Architecture

Based on the exposition on Section 4.1, the general idea of our SE(3)-equivariant architecture, given an input signal $f : \Omega \to \mathbb{R}$ where $\Omega \subseteq \mathbb{R}^3$ is a three-dimensional grid, is to first compute the matrices $P : \Omega \to SO(3)$ of the moving frame diagonalizing $H_{\sigma'} f(\mathbf{x})$ for every $\mathbf{x}$, i.e., find $P(\mathbf{x})$ such that $P(\mathbf{x})H_{\sigma'} f(\mathbf{x})P(\mathbf{x})^T$. Even if all eigenvalues are different there are at least two choices of normalized eigenvectors (i.e. columns of $P(\mathbf{x})$) corresponding to each eigenvalue, therefore to remove ambiguity an keep a consistent moving frame, we choose the option that has smallest angle with the gradient, and if the gradient norm is too small we multiply that column by zero.

After computing $P$ we compute blocks as shown in Figure 2$(b)$, which we call SE3MovF blocks, using the moving frame and the current features maps as input. The scale of each

layer does not need to be necessarily the same, here we consider that a scale $\sigma'$ is used to compute the moving frames and a scale $\sigma$ to compute the derivatives at each block.

A simple form of a global architecture, which we call SE3MovFNet, is in Figure $2(a)$. The feature maps of each block are summed like in residual networks, which mimics a PDE scheme (Ruthotto and Haber, 2020). The network in Figure 2 is specialized for a fixed number of channels, but by applying an $1 \times 1 \times 1$ convolution between blocks we can increase the number of feature maps of the next layer. Pooling may also be performed by subsampling after a block. The global max-pooling at the end renders the equivariant architecture invariant (Bronstein et al., 2021), which is interesting for a classification problem.

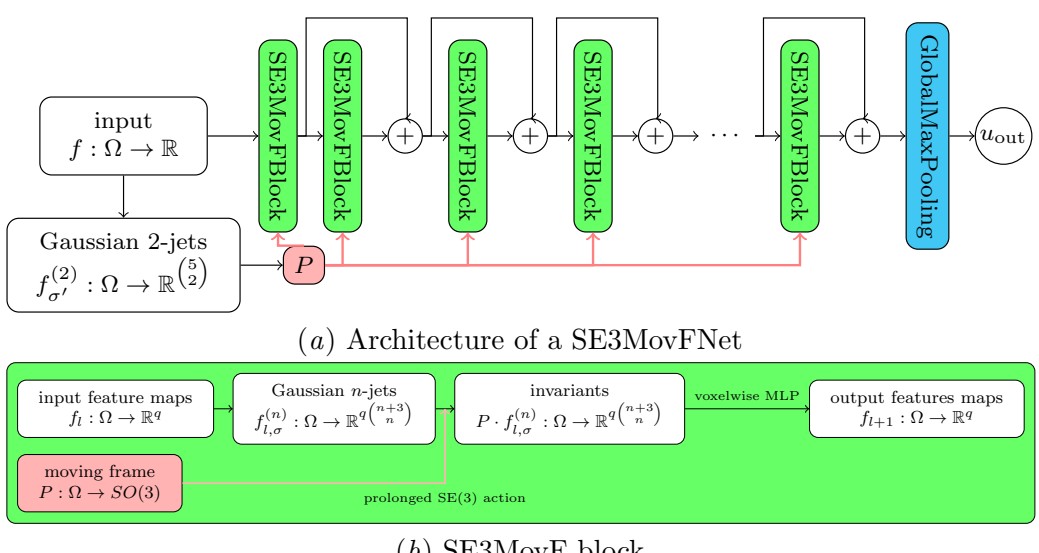

$(a)$ Architecture of a SE3MovFNet

$(b)$ SE3MovF block

Figure 2: An example of a SE3MovFNet architecture for classification, along with its fundamental build block the SE(3) MovF block. The block $P$ denotes the computation of the moving frame matrices $P : \Omega \to SO(3)$.

## 6. Experiments

**MedMNIST.** MedMNIST (Yang et al., 2021a,b) is a collection of datasets for benchmarking algorithms in medical image processings classification tasks. It contains six datasets of $28 \times 28 \times 28$ volumes: AdrenalMNIST3D, NoduleMNIST3D, VesselMNIST3D, SynapseMNIST3D, OrganMNIST3D, FractureMNIST3D. For more information see Yang et al. (2021b).

For each dataset we train a network with five SE3MovFr blocks with $16, 16, 32, 32, 64$ filters, using a stride of two in the second block. Voxelwise MLPs are computed as two subsequent $1 \times 1 \times 1$ convolutions followed by batch normalization (both) and leaky ReLU (only the first) and with the same number of neurons. We also train a CNN baseline with the same number of filters where each block consists of two $3 \times 3 \times 3$ convolutions followed by batch normalization and leaky ReLU. Input volumes are resized to $29 \times 29 \times 29$ so that

Table 1: Accuracies on the MedMNIST dataset compared to the benchmarks.

| | OrganMNIST3D | NoduleMNIST3D | FractureMNIST3D | AdrenalMNIST3D | VesselMNIST3D | SynapseMNIST3D |
|---|---|---|---|---|---|---|
| ResNet18 + 3D (Yang et al., 2021b) | 0.907 | 0.844 | 0.508 | 0.721 | 0.877 | 0.745 |
| ResNet18 + ACS (Yang et al., 2021b) | 0.900 | 0.847 | 0.497 | 0.754 | 0.928 | 0.722 |
| ResNet50 + 3D (Yang et al., 2021b) | 0.883 | 0.847 | 0.484 | 0.745 | 0.918 | 0.795 |
| ResNet50 + ACS (Yang et al., 2021b) | 0.889 | 0.841 | 0.517 | 0.758 | 0.858 | 0.709 |
| auto-sklearn (Yang et al., 2021a) | 0.814 | **0.914** | 0.453 | 0.802 | 0.915 | 0.730 |
| 3DMedPT (Yu et al., 2021) | - | - | - | 0.791 | - | - |
| CNN baseline (ours) | **0.927** | 0.871 | 0.528 | 0.824 | 0.949 | 0.775 |
| SE3MovFrNet (ours) | 0.745 | 0.871 | 0.615 | 0.815 | 0.953 | **0.896** |
| CNN baseline, augmented (ours) | 0.602 | 0.856 | 0.564 | 0.820 | 0.933 | 0.803 |
| SE3MovFrNet, augmented (ours) | 0.756 | 0.875 | **0.636** | **0.830** | **0.958** | 0.894 |

subsampling by a factor of two is equivariant by rotations of 90° around the coordinate-axes. Overall results can be seen in Table 1. There we can see that the SE3MovFNet surpassed most of the benchmarks. Results of testing the models on rotated test sets are seen in Figure 3 and Appendix E. In those results we observe it has perfect invariance for 90° rotations, evidenced by the periodicity of results, and a generally better equivariance than the CNN baseline with or without augmentation. It suffers, however, a significant loss for orientations not multiple of 90°.

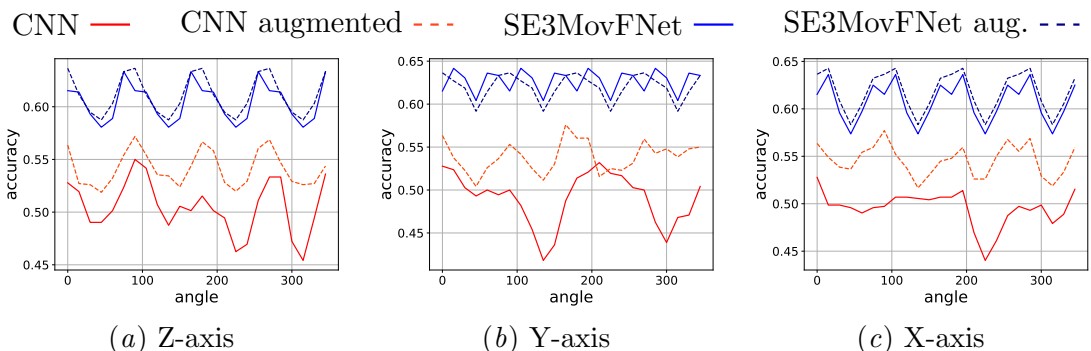

Figure 3: Predictions on the FractureMNIST3D on the test set rotated by different angles around each of the coordinate axis.

## 7. Conclusions

We have developed and successfully applied a SE(3)-equivariant architecture, SE3MovFNet, for a classification task in medical image processing. The proposed SE3MovFNet is an extension of a previous approach for SE(2)-equivariant networks (Sangalli et al., 2022) that corrects some of its numerical issues. The performance of our network is overall positive, as it attained the best results in 4 of the 6 evaluated datasets of MedMNIST and maintains a reasonable accuracy when images are rotated. Future work will explore other symmetries on other manifolds like for example scale and rotation symmetry simultaneously for images or volumes.

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

## Appendix A. Properties of Group Actions

If $\mathcal{G}$ is a Lie group acting on a manifold $\mathcal{M}$, we define the orbit passing through a point $\mathbf{z} \in \mathcal{M}$ as $\mathcal{O}_{\mathbf{z}}$ as the submanifold

$$\mathcal{O}_{\mathbf{z}} = \{g \cdot \mathbf{z} | g \in \mathcal{G}\}. \tag{15}$$

We can classify the action of a Lie group as

- *semi-regular* if all the group orbits have the same dimension;

- *regular* if it is semi-regular and each point $\mathbf{z} \in \mathcal{M}$ as an arbitrarily small neighborhood $U$ containing $\mathbf{z}$ such that the intersection of $U$ and each orbit is connected;

- *free* if for $g \in \mathcal{G}$, $\mathbf{z} \in \mathcal{M}$, $g \cdot \mathbf{z} = \mathbf{z} \implies g = e$ where $e$ is the identity on $\mathcal{G}$.

- *locally free* if there exists a neighborhood $U \subseteq \mathcal{G}$ containing $e$ satisfying $\forall g \in U$, $\mathbf{z} \in \mathcal{M}$ $g \cdot \mathbf{z} = \mathbf{z} \implies g = e$.

## Appendix B. From Invariants to Equivariant Operators

We assume we are in the context where we compute the invariantization of operators $F :$ $\mathcal{M}_1 \to \mathcal{M}_2$, with $\mathcal{M}_1 = X \times Y$ and $\mathcal{M}_2 = Z$, and the action $\mathcal{G}$ can be decomposed into actions in $X$ and $Y$, i.e.

$$g \cdot (\mathbf{x}, u) = (g \cdot \mathbf{x}, g \cdot u), \tag{16}$$

for all $g \in \mathcal{G}$, $(\mathbf{x}, u) \in X \times Y$. An equivariant operator in $\mathcal{M}_1$ can be related to an equivariant one in the space of functions $Y^X$. Suppose $\psi : \mathcal{M}_1 = X \times Y \to Z$ is $\mathcal{G}$-invariant, then take $\bar{\psi} : Y^X \to Z^X$ to be $\bar{\psi}[f](\mathbf{x}) = \psi(\mathbf{x}, f(\mathbf{x}))$, for all $\mathbf{x} \in X$, $f \in Y^X$. Assuming that the action on $Y$ is the identity $g \cdot u = u$ for $u \in Y$ and the same for $Z$ we have

$$
\begin{aligned}
\bar{\psi}(g \cdot f)(\mathbf{x}) &= \psi(\mathbf{x}, (g \cdot f)(\mathbf{x}))) \\
&= \psi(\mathbf{x}, f(g^{-1} \cdot \mathbf{x})] \\
&= \psi(g^{-1}\mathbf{x}, f(g^{-1} \cdot \mathbf{x}) \\
&= \bar{\psi}(f)(g^{-1} \cdot \mathbf{x}) = [g \cdot \bar{\psi}(f)](\mathbf{x}),
\end{aligned}
\tag{17}
$$

therefore $g \cdot \bar{\psi}(f) = \bar{\psi}(g \cdot f)$ for all $g \in \mathcal{G}$, $f \in Y^X$. In other words, an invariant operator in the Cartesian product $X \times Y$ to $Z$ induces an equivariant operator taking functions in $Y^X$ to functions in $Z^X$.

The same reasoning can be applied if $\psi : J^n(\mathcal{M}_1) = X \times Y^{(n)}$ is an invariant on the Jet-space and $\bar{\psi} : C^\infty(X, Y) \to C^\infty(X, Z)$ is the operator $\bar{\psi}[f](\mathbf{x}) = \psi(\mathbf{x}, \mathcal{J}_{\mathbf{x}}^{(n)} f)$ to show that $\bar{\psi}$ is $\mathcal{G}$-equivariant. We have

$$
\begin{aligned}
\bar{\psi}(g \cdot f)(\mathbf{x}) &= \psi(\mathbf{x}, \mathcal{J}_{\mathbf{x}}^{(n)}(g \cdot f)) \\
&= \psi(g^{-1} \cdot (\mathbf{x}, \mathcal{J}_{\mathbf{x}}^{(n)}(g \cdot f))) \text{ by invariance of } \psi \\
&= \psi(g^{-1} \cdot \mathbf{x}, \mathcal{J}_{g^{-1} \cdot \mathbf{x}}^{(n)}(g^{-1} \cdot g \cdot f)) \text{ by (4)} \\
&= \psi(g^{-1} \cdot \mathbf{x}, \mathcal{J}_{g^{-1} \cdot \mathbf{x}}^{(n)} f) \\
&= \bar{\psi}(f)(g^{-1} \cdot \mathbf{x}) = [g \cdot \bar{\psi}(f)](\mathbf{x}),
\end{aligned}
\tag{18}
$$

## Appendix C. Proof of Proposition 1

**Proof** For $l = 1$ we have

$$\phi_1(g \cdot f)(\mathbf{x}) = \psi_1 \left( \rho(\mathbf{x}, \mathcal{J}_{\mathbf{x}}^{(n)}(g \cdot f)) \cdot (\mathbf{x}, \mathcal{J}_{\mathbf{x}}^{(n)}(g \cdot f)) \right).$$

Noting $\mathbf{y} = g^{-1} \cdot \mathbf{x}$ and recalling (7), $g \cdot (\mathbf{x}, \mathcal{J}_{\mathbf{x}}^{(n)} f) = (g \cdot \mathbf{x}, \mathcal{J}_{(g \cdot \mathbf{x})}^{(n)} (g \cdot f))$, we can simplify

$$(\mathbf{x}, \mathcal{J}_{\mathbf{x}}^{(n)}(g \cdot f)) = (g \cdot \mathbf{y}, \mathcal{J}_{(g \cdot \mathbf{y})}^{(n)}(g \cdot f)) = g \cdot (\mathbf{y}, \mathcal{J}_{\mathbf{y}}^{(n)} f) = g \cdot \mathbf{z}_{\mathbf{y}}^0. \tag{19}$$

Hence,

$$\begin{aligned}
\phi_1(g \cdot f)(\mathbf{x}) &= \psi_1\left(\rho(g \cdot \mathbf{z}_{\mathbf{y}}^0) \cdot (g \cdot \mathbf{z}_{\mathbf{y}}^0)\right) \\
&= \psi_1\left(\rho(\mathbf{z}_{\mathbf{y}}^0) \cdot \mathbf{z}_{\mathbf{y}}^0\right) \quad \text{by invariance of } \mathbf{z} \mapsto \rho(\mathbf{z}) \cdot \mathbf{z} \\
&= \phi_1(f)(\mathbf{y}) \quad \text{by (11)} \\
&= \phi_1(f)(g^{-1} \cdot \mathbf{x}) \\
&= (g \cdot \phi_1(f))(\mathbf{x}) \quad \text{by definition of the action on functions.}
\end{aligned} \tag{20}$$

Therefore, $\phi_1(g \cdot f) = g \cdot \phi_1(f)$.

Now, for $l > 1$, we have for the case (12),

$$\phi_l(g \cdot f)(\mathbf{x}) = \psi_l\left(\rho(\mathbf{x}, \mathcal{J}_{\mathbf{x}}^{(n)}(g \cdot f)) \cdot (\mathbf{x}, \mathcal{J}_{\mathbf{x}}^{(n)} \phi_{l-1}(g \cdot f)_j)_{1 \le j \le q_l}\right). \tag{21}$$

As shown earlier, $(\mathbf{x}, \mathcal{J}_{\mathbf{x}}^{(n)}(g \cdot f)) = g \cdot \mathbf{z}_{\mathbf{y}}^0$ with $\mathbf{y} = g^{-1} \cdot \mathbf{x}$. Similarly, assuming that $\phi_{l-1}$ is equivariant,

$$(\mathbf{x}, \mathcal{J}_{\mathbf{x}}^{(n)} \phi_{l-1}(g \cdot f)_j) = g \cdot (\mathbf{y}, \mathcal{J}_{\mathbf{y}}^{(n)} \phi_{l-1}(f)_j)) = g \cdot \mathbf{z}_{\mathbf{y}}^{l-1,j}, \tag{22}$$

so that

$$\phi_l(g \cdot f)(\mathbf{x}) = \psi_l\left(\rho(g \cdot \mathbf{z}_{\mathbf{y}}^0) \cdot (g \cdot \mathbf{z}_{\mathbf{y}}^{l-1,j})_{1 \le j \le q_l}\right). \tag{23}$$

Since furthermore $\rho(g \cdot \mathbf{z}_{\mathbf{y}}^0) = \rho(\mathbf{z}_{\mathbf{y}}^0) \cdot g^{-1}$ by definition of a moving frame, we finally get

$$\begin{aligned}
\phi_l(g \cdot f)(\mathbf{x}) &= \psi_l\left(\rho(\mathbf{z}_{\mathbf{y}}^0) \cdot g^{-1} \cdot (g \cdot \mathbf{z}_{\mathbf{y}}^{l-1,j})_{1 \le j \le q_l}\right) \\
&= \psi_l\left(\rho(\mathbf{z}_{\mathbf{y}}^0) \cdot (\mathbf{z}_{\mathbf{y}}^{l-1,j})_{1 \le j \le q_l}\right) \\
&= \phi_l(f)(\mathbf{y}) = \phi_l(f)(g^{-1} \cdot \mathbf{x}) \\
&= (g \cdot \phi_l(f))(\mathbf{x}).
\end{aligned} \tag{24}$$

As for the case (13),

$$\begin{aligned}
\phi_l(g \cdot f)(\mathbf{x}) &= \phi_{l-1}(g \cdot f)(\mathbf{x}) + \psi_l\left(\rho(\mathbf{x}, \mathcal{J}_{\mathbf{x}}^{(n)}(g \cdot f)) \cdot \mathcal{J}_{\mathbf{x}}^{(n)} \phi_{l-1}(g \cdot f)\right) \\
&= \phi_{l-1}(f)(\mathbf{y}) + \psi_l\left(\rho(\mathbf{z}_{\mathbf{y}}^0) \cdot \mathbf{z}_{\mathbf{y}}^{l-1}\right) \quad \text{like above, with } \mathbf{y} = g^{-1} \cdot \mathbf{x} \\
&= \phi_l(f)(\mathbf{y}) \\
&= (g \cdot \phi_l(f))(\mathbf{x}).
\end{aligned} \tag{25}$$

Therefore in all cases $\phi_l(g \cdot f) = g \cdot \phi_l(f)$ provided this is true for $\phi_{l-1}$, and the proposition follows by induction. ∎

## Appendix D. Complexity Analysis

Let us assume that the input is given as a signal $f : \Omega \to \mathbb{R}$ where $\Omega = \{0, \ldots, W - 1\} \times \{0, \ldots, H - 1\} \times \{0, \ldots, D - 1\}$ is a grid of size $W \times H \times D$. Moreover, let us assume that we compute the moving frame using Gaussian derivatives of scale $\sigma'$ and the derivatives at other layers using $\sigma$, and that the discrete Gaussian derivative filters all have dimension $w' \times w' \times w'$, for the moving frame and $w \times w \times w$ for the other layers.

The computation of the moving frame is done as follows:

- compute all Gaussian derivatives of order one and two of $f$. Gaussian derivatives are separable, thus each one can be obtained by three convolutions with a filter of size $w$, which have cost a cost of $O(WHDw')$ floating point operations (flops);

- compute the eigenvectors of the Hessian. Since the matrices have constant size $3 \times 3$ we consider this operation is done in constant time for each pixel and this step is done in $O(WHD)$ flops.

So the computation of the moving frame is done in $O(WHDw')$ flops.

From there on if we compute a layer with $q'$ input feature maps and $q$ output feature maps:

- this layer computes $\binom{n+3}{n}$ Gaussian derivatives for each input feature map, where $n$ is the order of differentiation used, resulting in $O\left(\binom{n+3}{n} q' w WHD\right)$ flops;

- to compute the prolonged group action, it can be verified that the equivariant group action can be expressed as polynomial in the partial derivatives, and thus it takes $O\left(\binom{n+3}{n} q' WHD\right)$ flops;

- the previous step is followed by an $L$-layer multi-layer perceptron at each voxel. Assuming that the output dimension at each layer of the MLP is at most $q$ we have that this step takes $O(q'q + Lq^2)$ flops.

The complexity of a layer of SE3MovF is the sum of the complexity of each step, i.e. it can be done in $O\left(\binom{n+3}{n} q' w WHD + q'q + Lq^2\right)$ flops. In our experiments here we used $n = 2$ and $L = 2$ for all models, so the impact of those terms is very limited.

## Appendix E. Additional Results

Figures 4 and 5 show some more results on MedMNIST3D.

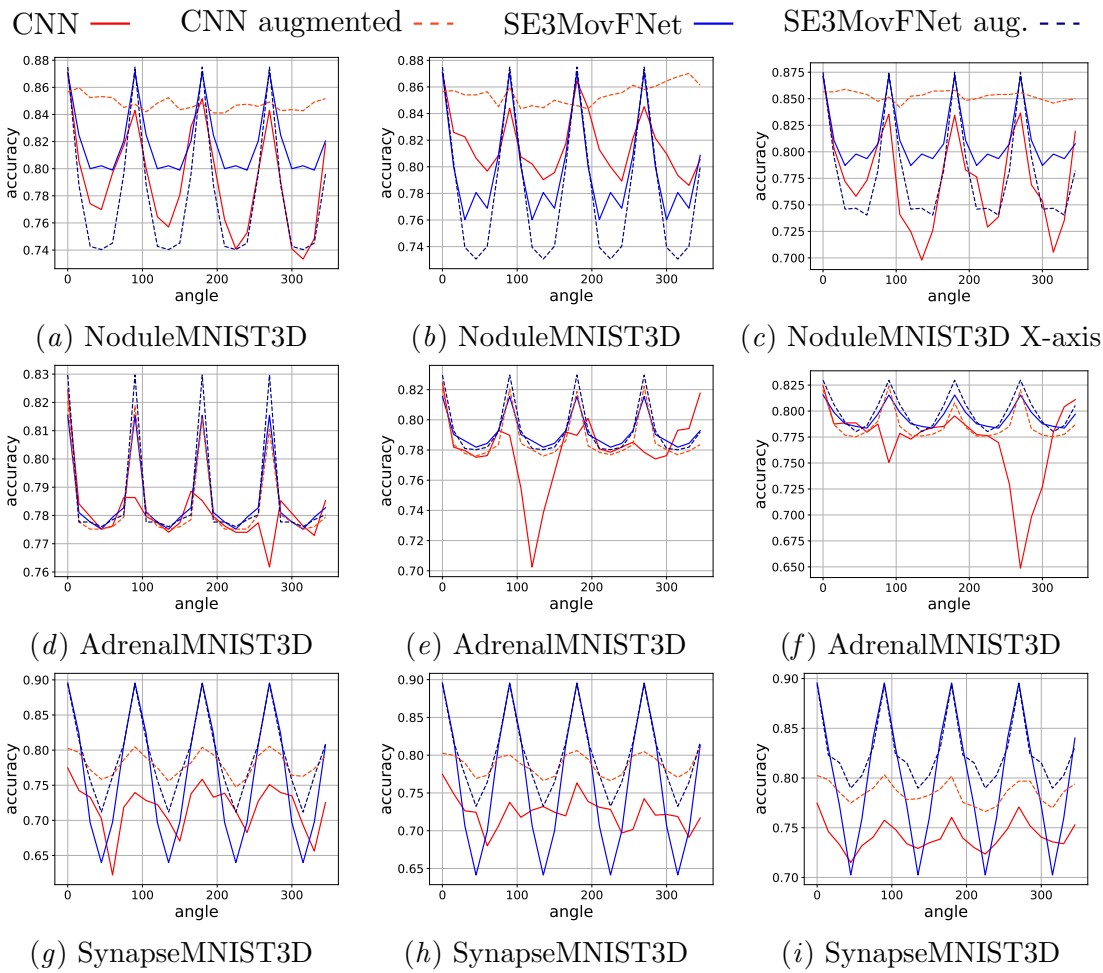

Figure 4: Additional results of evaluation on rotated volumes on the coordinate axes for NoduleMNIST3D, AdrenalMNIST3D and SynapseMNIST3D. In the first column volumes are rotated around the Z-axis, In the second Y-axis and in the third column X-axis.

.

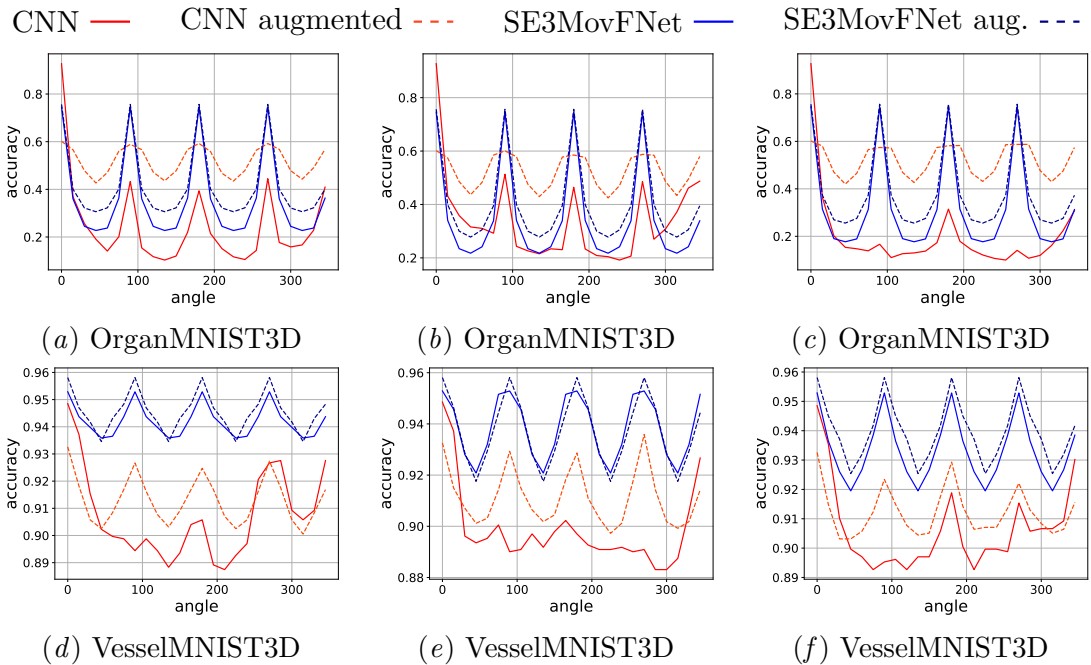

Figure 5: Additional results of evaluation on rotated volumes on the coordinate axes for Organ­MNIST3D, VesselMNIST3D. In the first column volumes are rotated around the Z-axis, In the second Y-axis and in the third column X-axis.

