# OpenReview forum: "Moving Frame Net: SE(3)-Equivariant Network for Volumes"
_NeurIPS.cc/2022/Workshop/NeurReps — NeurReps 2022 Poster_

### Official Review · Reviewer_LMGr · 2022-10-13
**Extension of SE(2) equivariant nets to SE(3) with moving frames**

**Confidence:** 4
**Soundness:** 3
**Presentation:** 3
**Contribution:** 2
**Overall Rating:** 5

**Summary:**

This paper builds on the derivation of cross-sections and moving frames to derive an SE(3)-equivariant neural network for 3d medical image classifications. A short review of the literature on group-equivariant networks is first exposed. A distinction between networks for point-clouds and images/volumetric data is made, the latter case being developed in this paper.
Then the technical background is recalled, with definitions of group actions, invariance and equivariant properties and moving frames. The key is that finding a cross-section defines a moving frame, which in turn defines a invariant map. Jet bundles are used to prolongate the group action in cases where it is not free.
These notions are then made explicit in the case of SE(3), and the method to derive an equivariant network is proposed, with the specificity that only the moving frame of the input is used throughout the network. It is proved that the resulting layers remain equivariant maps and applies to 3d data after applying a Gaussian filter.
The proposed architecture is tested on 6 data sets and compared against a vanilla CNN with the same number of layers and feature maps. The proposed model is said to outperform benchmarks and an analysis of its performance against rotations of the input show that it is robust to 90° rotations.
Finally a short conclusion recalls the main contribution of the paper.

**Questions:**

How many samples contain the datasets? Are the results significant?
Can the code be made publicly available?
Shouldn't SE(3) be defined as a semi-direct product and not a direct product in section 4?
In the definition of the cross section, the eigenvalues are placed in non-decreasing order aren't they?
There are few typos in the last paragraph of section 4: the ? of G, of of, can be seen as input is repeated twice...
Throughout the paper: use "an" instead of "a" in front of a letter: an n-dim manifold, an MLP, an SE(2)-equivariant...


**Limitations:**

Limitations of the proposed model are not discussed.

**Recommended Decision:**

2: Borderline

**Relevance:**

3: Solid fit

**Strengths And Weaknesses:**

The paper is generally well written and clear.
Its originality lies in the extension of existing architectures on SE(2) to SE(3), and a major numerical simplification that allows to perform the computations in this case, with a proof that the desired properties remain valid.
The paper is technically sound and provides a good mathematical treatment of the proposed method.
The literature review could be more detailed, for example, works by Erik Bekkers et al, Fabian Fuchs or Yan Li and colleagues also proposed (SE(2-3)) group-equivariant networks and applied it to medical image analysis.
The results of the experiment are actually significant on 2 of the 6 datasets, more experiments should be performed to demonstrate the improvement of the method.

**Submission Track:**

Proceedings Paper (9 Page)

---

> ### Author Response · Authors · 2022-10-21
> **Answers to questions**
>
> Thanks for the suggestions of references.
>
> > How many samples contain the datasets?
>
> In average the datasets have around 1500 samples each, with a ratio of around 70% training / 10% validation / 20% test.
> More details can be found in the second table here https://medmnist.com/
>
> > Are the results significant?
>
> We believe the results on the FractureMNIST3D and SynapseMNIST3D datasets to show a significant improvement over the previous state-of-the-art.
>
> > Can the code be made publicly available?
>
> Yes, we intend put it on github by the time of the workshop
>
> > In the definition of the cross section, the eigenvalues are placed in non-decreasing order aren't they?
>
> It is in non-increasing order from top-left to bottom-right.

---

### Official Review · Reviewer_iFCj · 2022-10-14

**Confidence:** 2
**Soundness:** 3
**Presentation:** 2
**Contribution:** 3
**Overall Rating:** 4

**Summary:**

The authors propose a novel SE(3)-equivariant voxel-based model, which is based on the theory of moving frames. It is an alternative to previously proposed equivariant voxel-based approaches, which use, for example, the spherical harmonics. The proposed method is currently limited to SE(3)-invariant classification problems, but the authors state than an extension to general SE(3)-equivariant voxel-based problems is possible. The method is evaluated on the MedMNIST datasets and is shown to be robust to testing set transformations.

**Questions:**

* What steps would need to be taken to apply the proposed method to voxel-based semantic segmentation.
* Is it possible to compare spherical harmonics based SE(3)-equivariant methods on MedMNIST?
* An important hyper-parameter in the spherical harmonics based methods is the maximum degree of the harmonics. Is there an analogous hyper-parameter in your method?

**Limitations:**

I would like to see discussion of computation and memory costs, training time, network capacity and applicability beyond SE(3)-invariant classification problems.

**Recommended Decision:**

1: Reject

**Relevance:**

4: Highly relevant

**Strengths And Weaknesses:**

## Strengths:
* The authors propose a novel method for creating SE(3)-equivariant voxel-based neural architecture through the application of the theory of moving frames. While currently limited, a future version of this paper could create a compelling alternative to methods based on spherical harmonics.
* The proposed method reaches state-of-the-art results on the MedMNIST dataset and is robust to some transformations of the testing set.

## Weaknesses:
* Only one dataset, MedMNIST, is used in this paper. I am unfamiliar with the dataset and I would like to hear from the other reviewers if they find it sufficient. Usually, voxel-based methods (e.g. https://ieeexplore.ieee.org/document/7353481, https://arxiv.org/abs/1611.08974) are evaluated on voxelized point cloud datasets.
* None of the baselines used are SE(3)-equivariant, despite the authors pointing out several SE(3)-equivariant voxel-based methods in the related work section.
* The benefits and drawbacks of achieving SE(3)-equivariance through spherical harmonics versus moving frames is not discussed. In particular, Figure 2 shows that the proposed method deteriorates when the input is not rotated by multiples of 45 or 90 degrees. I presume spherical harmonics do not have this limitation.
* Only an invariant version of the method for classification is demonstrated. The paper could be greatly strengthened by creating an equivariant version for tasks such as voxel-based semantic segmentation (e.g. https://arxiv.org/abs/1611.08974).

## Comments
* The last paragraph of Section 3.1 is unclear. I do not know what the authors mean by “we do not consider actions that change the output of the function f”. In the case of equivariance, we are acting on both the input and the output domain. Similarly, in the sentence “As this paper is focused on exploring equivariant networks on signals, this is the type of action we seek equivariance to.”, I am not sure what the second occurrence of “this” refers to.
* The theoretical background in Section 3 and the construction of the method in Section 4 are unapproachable. I understand that the underlying theory is complex, but comparable works (e.g. https://arxiv.org/abs/1802.08219) spend a lot of effort to crystallize the core equivations and the intuitions behind them. Additional figures would be helpful.
* Section 4: “[...], but keeping in mind that for higher dimensions it is just a matter of channel-wise application.” Channel-wise application of ___?
* In Table 1 – NoduleMNIST3D – auto-sklearn, the wrong number is copied from https://arxiv.org/abs/2110.14795.
Related to above, please report both AUC and Accuracy, as in the original MedMNIST paper. Please also report results for multiple random seeds.


**Submission Track:**

Proceedings Paper (9 Page)

---

> ### Author Response · Authors · 2022-10-21
> **Answers to questions**
>
> > "What steps would need to be taken to apply the proposed method to voxel-based semantic segmentation."
>
> That would require not using a pooling as the last layer.
> In the present scenario (classification) we computer layers $\phi_1, ... \phi_L$ as detailed in Proposition 1, and we apply a pooling layer (global max pooling or global average pooling) to $\phi_L$, followed by a fully connected layer, to obtain the classification logits.
> If the goal was to perform segmentation we could apply a $1 \times 1$ convolution layer with the number of classes as the number of feature maps and that would define the pixel-wise logits.
>
> > "Is it possible to compare spherical harmonics based SE(3)-equivariant methods on MedMNIST?"
>
> Yes, the code for spherical harmonics SE(3)-equivariant methods are available online and so we should be able to test it in MedMNIST. Thanks for the suggestion, that would be an interesting comparison.
>
> > "An important hyper-parameter in the spherical harmonics based methods is the maximum degree of the harmonics. Is there an analogous hyper-parameter in your method?"
>
> In our method, the closest hyperparameter to the degree of the harmonics would be the orders of the Gaussian derivatives. (not the derivatives used to compute the moving frame though, for those we only need order 2)
> However we do not need to keep track of orders of the derivatives in the following layers.

---

### Official Review · Reviewer_RQWR · 2022-10-14
**Review for "Moving Frame Net: SE(3)-Equivariant Network for Volumes"**

**Confidence:** 3
**Soundness:** 3
**Presentation:** 3
**Contribution:** 3
**Overall Rating:** 7

**Summary:**

The authors present a method for rotation and translation equivariant neural networks for voxel data, and evaluate their proposed method thoroughly in a variety of experiments.

**Questions:**

Some of the mathematical details in Section 4 (such as the prolongation of the action to a second-order jet space, and the proof of Proposition 1) were a bit difficult for me to understand using just the mathematical tools introduced in the earlier sections. I understand that it is difficult to introduce such a complicated topic within the provided page limit, but I wonder whether there is anything that can be done to make this a bit easier to understand.


**Limitations:**

The article does not really discuss its limitations in a thorough way. I believe that a limitations section after the experiments would improve this article significantly.

**Recommended Decision:**

3: Accept

**Relevance:**

4: Highly relevant

**Strengths And Weaknesses:**

The article introduces a good idea, and explains its mathematics well. I believe it should be accepted as a proceeding.

I enjoyed the mathematical background provided in Section 3, which made the article easy to understand and also educational in general.
The mathematics is, however, a little bit heavy for a general geometric machine learning audience. I suggest adding a few small illustrations of definitions to Section 3 to help with that issue.

The experiments and evaluations are good.

Minor language issues:
- Section 3.2: The sentence "A moving frame is a map G-equivariant rho [...]" is difficult to parse.

**Submission Track:**

Proceedings Paper (9 Page)

---

> ### Author Response · Authors · 2022-10-21
> **Answers to questions**
>
> > Some of the mathematical details in Section 4 (such as the prolongation of the action to a second-order jet space, and the proof of Proposition 1) were a bit difficult for me to understand using just the mathematical tools introduced in the earlier sections. I understand that it is difficult to introduce such a complicated topic within the provided page limit, but I wonder whether there is anything that can be done to make this a bit easier to understand.
>
> Thanks for pointing it out.
> I want to point out that there was small mistake in the definition of the prolonged group action, indeed it should be defined as
> $$
> g \cdot (\mathbf{x}, u^{(n)}) \coloneqq (g \cdot \mathbf{x}, \mathcal{J}_{(g \cdot u)}^{(n)}{g \cdot \mathbf{x}})
> $$
> I hope that the correct definition might make things clearer.

---

### Decision · Program_Chairs · 2022-10-21

Accept (Poster)